# Narrative in Older People Care—Concepts and Issues in Patients with Dementia

**DOI:** 10.3390/healthcare10050889

**Published:** 2022-05-12

**Authors:** Grażyna Puto, Patrycja Zurzycka, Zofia Musiał, Marta Muszalik

**Affiliations:** 1Department of Internal and Environmental Nursing, Institute of Nursing and Midwifery, Faculty of Health Sciences, Jagiellonian University Medical College, 31-501 Cracow, Poland; patrycja.zurzycka@uj.edu.pl (P.Z.); zofia.musial@uj.edu.pl (Z.M.); 2Department of Geriatrics, Collegium Medicum in Bydgoszcz, Nicolaus Copernicus University in Torun, 87-100 Torun, Poland; muszalik@cm.umk.pl

**Keywords:** narrative, older people, long-term care, narrative medicine, identity

## Abstract

Medical sciences in their classic approach focus on objectively measured dimensions of human functioning and its disorders. Therefore, they are often far removed from the unique identity, experiences and needs of older people. The solution to this type of focusing on the biological, psychological or social dimension of the life of older people may be the inclusion of the narrative in the daily practice of medical care. Narrative medicine supports the development of a holistic approach to care that allows older people to present their own life story, which helps to recognize their uniqueness and to show a genuine interest in the narrative. Attention is increasingly drawn to the fact that the narrative of older people should be recognized and taken into account when planning and providing care in institutions, including long-term care facilities (LTCFs). Despite the fact that LTCFs are often attended by people with multiple diseases and with cognitive impairment, the recognition, respect and maintenance of personal identity should constitute the foundation of caring activities. The basic premise of narration is the recognition that the development of identity does not stop at any age but continues throughout life, and that narrative is an important form of self-expression. The aim of this paper is to present selected issues related to the practice of narrative medicine in caring for older people.

## 1. Introduction

A classic understanding of medical sciences focuses on objectively measurable aspects of the functioning of an older person by perceiving the body and its dysfunctions in a way that is not linked in any way to that person’s life story, background, needs and uniqueness. On the other hand, from the perspective of narrative medicine, an older person and the disease are perceived as a complex phenomenon involving biological, psychological and social factors [1]. Narrative medicine—as defined by Rita Charon—is based on the ability of healthcare professionals to recognize, adapt, interpret and react to the stories of other people [2,3,4]. In order to be able to provide care in accordance with the principles of narrative medicine, it is necessary for healthcare professionals to possess competencies that involve communication skills, empathy and trust. An important role is also played by the personal reflection made by both health care workers (HCWs) and other people involved in the care [2,5,6].

Incorporating the narrative into daily medical care is about enabling older people to tell their own story, recognizing its uniqueness and showing a sincere interest in the story. It should be emphasised that the narrative concerns both past and current events that may be related to the patient–health care worker interaction [7,8,9,10]. The narrative is an essential tool that serves to integrate the different points of view of those involved in the therapeutic process, and its utmost priority is to develop a personalised treatment and care plan. The dialogue between older people, their relatives and health care professionals provides an opportunity to collect and analyse the information that allows one to see the perspective of all the participants involved in this interaction. It is essential that the emerging narrative should reflect a balanced contribution from all sides. If one version of the story becomes dominant, there is a risk that elements important to another contributor will be distorted. It is vital to avoid the domination of the biological, medical and clinical narratives focusing on pathologies understood as deficits occurring in older people, without taking into account the overall perspective of their functioning [11]. The result of practising narrative medicine is the integration of clinical evidence with the patient’s life story and the creation of a personalised care and treatment plan whose essence and purpose are seen from the perspective of the patient, not the medical staff or care institution. Therefore, narrative-based medicine (NBM) is combined with evidence-based medicine (EBM). Such a combination takes into account various perspectives and makes care-related decisions more personalised, effective and adequate to the needs of patients [1,7].

The aim of this paper is to present selected issues related to the practice of narrative medicine in caring for older people.

## 2. Narrative Medicine in the Care of Older People

Narrative medicine supports the development of a holistic approach to care not only in long-term care facilities, but also in the context of community care. The main idea behind community care is to provide integrated care based on the patient’s needs and multi-disciplinary assessment (e.g., medical and social care). Taking into account the background, experiences, observations and preferences of older people and their caregivers can reduce the risk of making wrong choices concerning care and therapeutic activities. By paying attention to the narrative of older people and their relatives, interdisciplinary therapeutic teams can interact based on the understanding of a human being as a complex system, rather than focusing on a single disorder or pathology [1,12,13].

The narrative approach involves both listening to and carefully following the stories told by older people, as well as paying close attention to the way in which they are expressed, and taking into account the circumstances in which the stories are told [14]. Every person’s life story is made up of countless experiences that accumulate over time—a life story is, therefore, constantly being written. Narratives shape who we are and who we become, thus influencing the continuous development of our identity. Telling a story, as well as repeating it, is essential for the development of the self, giving meaning to one’s existence and establishing relationships with the environment [12,13,14,15]. Narrative identity is a multidimensional phenomenon related to social, cultural and historical contexts as well as to time (the narrative develops in time and through time). In a certain sense, the past only exists as long as it is remembered and recreated in interaction with present and future experiences, as well as with the meanings, interpretations and metaphors connected to those experiences [14]. The basic assumption of narrative medicine is the recognition that the development of identity does not stop at any age, but continues throughout life, and that narration is an important form of expression of oneself [10,12,13,15,16,17,18].

Narrative care is the recognition of the “here and now”, i.e., the world jointly created and discovered by older people and the staff involved in narrative care. It is also a recognition that we all have a past and that we all have a future ahead of us [19]. The joint creation of the narrative by the patient and medical personnel is, therefore, a dynamic and continuous process that takes into account the past, the present and the future [11]. The development of a narrative is perceived as a potentially infinite process that is not subject to limitations due to age, health or emerging cognitive deficits [14,15,16,17,18]. Narrative gerontology unequivocally links ageing with the need for purposefulness, and in particular for a deeper and more satisfying sense of significance related to the individual uniqueness of a person’s life. The approach focuses both on the continuously created identity and the resulting need for attention. Narrative care involves recognizing and respecting the experiences and the life story of each person and—thanks to taking into account the resulting needs—plays a role as important as meeting that person’s biological needs (e.g., nutrition, self-care and treatment) [12,13,15,17].

The narratives cover both present and past events, as well as situations that never happened. They may relate to missed opportunities that are a source of the so-called “lost narratives”. Situations that have not been taken advantage of by a human being exert a significant impact on the personality, and thus shape the identity [14]. Narratives dealing with missed opportunities can be both an expression of regret and a desire for reconciliation, but they also provide a key that enables the listener to understand the person’s past life events. Looking at past events from the perspective of the present allows us to see new aspects that were not possible to grasp from the original perspective [14]. Although narrative care will not solve problems that reside in the past, it creates the possibility of gaining new insights about past events and thus may influence present and future events [19].

Personal narratives place themselves in the context of broader cultural narratives. Cultural narratives also interact with economic, institutional and political factors that influence the policies that affect ageing (including care and support organisations for older people) by imposing systemic narratives. Ageing is not a useless biological process that develops alongside chronological age, but a normative cultural construct. Some authors point out that the contemporary cultural narratives are stereotypical and consider the natural phenomenon of ageing unfavourable (negating old age as a natural stage of life). Such an approach favours the marginalisation of older people and their narrative, which may lead to exclusion or the feeling that life has nothing new to offer, and ultimately, to the belief that the time of development opportunities has come to an end, although life is not over yet [20,21,22,23].

## 3. Narrative in Long-Term Care of Older People Suffering from Dementia

An increasing amount of attention is being paid to how patient narratives could be incorporated into medical care in long-term facilities. In this form of care, it is claimed that recognizing, respecting and maintaining personal identity must be considered as the basis of caring practice [12].

Changing the place of residence of an older person, which results in a stay in a long-term care institution, can be perceived as an irreversible event related to the end of one’s life. This may lead to the belief that the life story of a given person is over and that there are no prospects for its further creation (narrative exclusion) [24]. Failure to respect the individual uniqueness of older people staying in a long-term care facility deprives them of control over their own lives (e.g., the possibility of building a narrative). This may result in passivity, confusion, irritation or aggression, which in the case of older people with cognitive impairment may be interpreted as a symptom of the disease [24].

Narrative care, involving interventions focused on older people and their unique life narrative, should not focus solely on existing functional deficits in activities of daily living (ADL). Thanks to this approach, people with cognitive disorders can regain their own voice, which is often overlooked and marginalised [25]. Recognition that older people, their relatives and the staff who care for them jointly create narratives is the first step towards implementing holistic care based on narrative practice [12,13]. Narrative care, which also requires the involvement of caregivers and the patient’s relatives [1], should also be practised during regular interactions between an older person and the employees of a long-term care facility (e.g., when dressing up or eating meals) [19,24]. It is worth noting that most of those types of relationships are narrative in their nature, although the people involved in a given interaction are not always aware of it [13].

Older people staying in long-term care facilities, due to their limitations in the ability to function independently, may be perceived as people who have lost their narrative abilities. This narrative exclusion is associated with a serious threat to the preservation of the narrative uniqueness, which results in the progressive loss of identity due to the severing of existing interpersonal ties, loss of habits and of the rhythm of everyday life that together constitute the essence of who an older person is, since these are the components of their stories [24,26,27]. The biomedical perception of dementia causes older people to be seen as incapable of telling their story in a coherent way, connecting past experiences with the present, updating the narrative or sharing experiences [19]. Maintaining the ability to express the narrative (preserving the narrative identity) may indeed be challenging, especially in the case of older people who suffer from dementia and whose abilities to recall memories and build narratives are at risk due to the progression of the disease, especially in its advanced stages [24]. Narrative agency can be significantly impaired in older people with dementia due to several factors. The nature of the disease makes it increasingly difficult to construct and articulate coherent and meaningful stories and to share them with others. On the other hand, the cultural view of dementia promotes a negative and pessimistic image of older people who suffer from this disorder. The convergence of these two factors can lead to the emergence of caring interactions that do not support any narrative activity and may even prevent older people with dementia from creating and expressing their stories. When dementia is perceived as the dominant trait, an older person loses their own subjectivity, and their identity comes to be defined solely by the disease and the importance others attach to it, which can influence the quality of interpersonal interactions. One should strive to minimise the association of older people with cognitive impairment only from the perspective of biomedical pathology, focusing on their deficits, while instead one should try to enhance the subjectivity and the unique narrative of a person—perceiving the person and not the patient whose behaviour is seen solely through the prism of the disease [25].

Depersonalising older people with dementia not only discourages communication and contributes to neglect, but also reduces the quality of the few existing interactions. Unfortunately, caregivers themselves may contribute to depersonalization by addressing older patients using diminutives, speaking very slowly, simplifying what they say, using incorrect pronouns, using exaggerated intonation or asking questions that clearly indicate the expected answer. This type of verbal communication, although motivated by good intentions, leads to infantilization in relationships with older patients, and may suggest their dependence and lack of competence. In long-term care facilities for older people, most of the interactions undertaken by staff are focused on instrumental tasks—assistance in performing personal activities of daily living (PADL) and medical tasks related to health control and implementation of the therapeutic process. Therefore, in most older people, the interaction consists mainly of making the resident cooperate during a certain activity. Therefore, the conversation initiated and directed by the carer is often only instructional, consisting of a set of short, standardised imperative and affirmative sentences, as well as comments whose aim is to evaluate the achieved progress of the task, with little possibility of active participation by the resident [24,25,28,29,30,31].

The narrative involves both verbal communication (storytelling) and non-verbal communication. Therefore, narrative care should be implemented regardless of the existing disorders and limitations in the possibilities of verbal communication. In the case of older people with communication disorders, it is the non-verbal expression (gestures, movements, facial expressions) that can be the main form of narration (both recreating it and constructing new stories in interaction with patients’ loved ones and the staff) [13,19]. First and foremost, strengthening the narrative of older people involves encouraging them to build it. In the case of patients who show disturbances in verbal communication, it is important to strengthen various forms and methods of self-expression—non-verbal communication, alternative methods of communication, but also the possibility of expression using music, dance or other forms of art, which also allow for the creation of stories and the expression of one’s identity. It is important for the caregivers to build a narrative space, i.e., to demonstrate genuine commitment and to find sufficient time and attention for the patients. Creating conditions conducive to creating narratives requires both individual effort and the right conditions related to the organisation of work in a given institution. This requires a departure from performing only task-focused care in favour of a wider consideration of interpersonal relationships [24], in which caregivers are the co-authors of the narratives created by the residents [13].

## 4. Conclusions

Carrying out nursing care that takes into account the narratives of older people is based on one of the basic assumptions of the nursing profession, which is being mindful of and attentive towards another human being. The narrative makes it possible to perceive an older person from a multidimensional perspective that takes into account both biological, psychological and social factors. Such a broad approach prevents excessive focus on the biomedical dimension of functioning and on the existing functional disabilities and diseases. Including the narrative in the care of older people is based on engaging them—in a manner adequate to their abilities and needs—in creating the narrative and in expressing their identity. Incorporating the narrative into the practice of providing medical services for older people in general and not limiting it to those who suffer from dementia has a significant impact on the possibility of providing holistic and individualised care, while at the same time taking into account the values considered important by older people. Efforts should be made to ensure that residents in institutional care are not deprived of the possibility of self-expression. Personal narratives should not be dominated by a biomedical perspective, where the ageing process is viewed as a gradual and irreversible loss of many abilities, including narrative-building.

Incorporating narrative into daily practice requires creating a narrative-friendly environment that enables storytelling while taking into account the caregivers’ skills and attitudes. This includes the acceptance of narration, the ability to listen actively, interpreting non-verbal language and taking into account individual and cultural conditions and organisational settings, as well as carefully analysing the received information and drawing conclusions. This also applies to the organizational structure, i.e., the need to abandon the practice of care based on tasks related to satisfying only biological necessities, approval of narrative practice at the level of the institution or the health care system of older people, including the patient’s relatives in the narratives, as well as counteracting narratives promoting a negative and pessimistic image of an older person. It also requires the staff to spend a significant amount of time with the residents and to show a genuine interest in them. It is also crucial to reflect on how the narrative affects older people’s needs and the care provided to them, as well as what impact it has on their own identity and their sense of connection to the outside world, and how it ultimately allows for holistic care that takes into account the patient’s specific situation.

## 5. Limitations

It should be emphasised that the present work has some limitations. Firstly, it is a communication presenting a number of definitions, pieces of information and conclusions. Secondly, in the review of the existing literature, due to the small number of publications on this subject, the specific types of the analysed studies were not taken into account (review, research).

The strength of the paper is that—in the context of the increasing proportion of older people in the population—the subject of narrative medicine, which is very rarely discussed, has gained a great deal of importance. Narrative medicine supports the development of a holistic approach to care, which consists of empowering older people to present their own life story, recognizing their uniqueness and showing a genuine interest in their narrative.

## Data Availability

Not applicable.

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
