# Peer review of "Narrative in Older People Care—Concepts and Issues in Patients with Dementia"

_healthcare, 2022, doi:10.3390/healthcare10050889_

Round 1
Reviewer 1 Report
Brief summary
This manuscript was designed with the aim to present on the selected issues related to the practice of narrative medicine in caring for the older people. The strength of this manuscript is the discussion the topic of narrative medicine, an area that has not been widely discussed as compared to the objective measurement of human functioning and disorders.
General comment
The author stated that the main objective of this paper was to present on the selected issues related to the practice of narrative medicine in caring for the older people However, it is quite difficult to grasp the issues to be discussed especially in the subtopic of 2. Narrative medicine in the care of older people. This paragraph is quite long and mainly discussing on the concept of the narrative medicine rather than any the presentation of issues related to the topic. In addition, the issues presented were mainly related to patients with dementia. I would suggest changing the title of the manuscript to something as : Narrative in the Older People Care – Concept and Issues in Patients with Dementia.
In the limitation section, the author mentioned that this manuscript is a systematic review. This seems inaccurate, because the design of this paper did not include any systematic way of searching article that complies with the PRISMA guideline, for example. Besides, there were unclear/absence of discussions on the knowledge gap or future research on the topic. I would suggest that the author addressed these components first if this manuscript is accepted for publication.
Conclusion: It would be good if the conclusion include the summary of the issues related to the practice of narrative medicine in caring for the older people on top of the concept.
Author Response
Dear Editor,
Dear Reviewers,
Thank you very much for all your comments and suggestions.
There are very few reports in the scientific literature about the narrative in the older people care. Most of the papers dealing with this issue are selective in their nature. Moreover, there are no articles that would provide information on the narrative in older people. The authors, with the aim to introduce this subject, gave a general presentation in: „2. Narrative medicine in the care of older people”
Thank you for the comment on the title - it has been changed.
Line 2; end citation: deleted “Narrative in the Older People Care – Selected Issues” added “Narrative in the Older People Care – Concept and Issues in Patients with Dementia”
Thank you for the helpful comment. The article is not a systematic review, it is a communication.
Line 244: deleted “systematic review” added “communication”
Line 229: added/ corrected
This includes the acceptance of narration, the ability to listen actively, interpreting non-verbal language and taking into account individual and cultural conditions, as well as carefully analyzing the received information and drawing conclusions. This also applies to the organizational structure, i.e. the need to abandon the practice of care based on tasks related to satisfying only biological necessities, approval of narrative practice at the level of the institution or the health care system of the older people, including the patient's relatives in the narratives, as well as counteracting narratives promoting a negative and pessimistic image of an older person. It also requires the staff to spend a lot of time with the residents and to show a genuine interest in them. It is also crucial to reflect on how the narrative affects older peoples’ needs and the care provided to them, as well as what impact it has on their own identity and their sense of connection to the outside world, and how it ultimately allows for holistic care that takes into account the patient’s specific situation.

Reviewer 2 Report
It has been a pleasure to read your article. I have only minor suggestion.

Author Response
Dear Editor,
Dear Reviewers,
Thank you very much for all your comments and suggestions.
Of course, the authors agree that the narrative incorporates both verbal communication (storytelling) and non-verbal communication, and the narrative care should be put into practice regardless of the existing disorders and limitations in the possibility of verbal communication.
Non-verbal communication in the narrative care of older people is important, however, in this particular part of the manuscript (Line: 187 - 192) the authors focused on verbal communication rather than non-verbal communication.
Best regards.
Grażyna Puto

Reviewer 3 Report
"Narrative in the Older People Care" is a very interesting topic. I suggest some revision for improving quality of study.
(Comment 1) Population aging is a global problem, and countries are promoting community care in their own way. I recommend authors to supplement relationship between narrative care and community care in Section 2 (line 62-121).
- The main concept of community care is to provide integrated care based on the patient's needs and multidisciplinary judgment (e.g. medical service and social service).
(Comment 2) The authors described narrative care for older people with dementia. Narrative care has a lot to do with the primary care who practices general medicine. I recommend authours to mention in the manuscript (e.g. limitation or conclusion) that narrative care can be applied to ‘providing medical services’ for the elderly, not limited to dementia.
Author Response
Dear Editor,
Dear Reviewers,
Thank you very much for all your comments and suggestions.
Comment 1
Line 65: added „not only in long-term care facilities, but also in the context of community care. The main idea behind community care is to provide integrated care based on the patient's needs and multi-disciplinary assessment (e.g. medical and social care)”.
Comment 2
Line 220: added “Incorporating the narrative into the practice of providing medical services for older people in general and not limiting it to those who suffer from dementia has a significant impact on the possibility of providing holistic and individualised care, while at the same time taking into account the values considered important by older people”.
Best regards.
Grażyna Puto
